# ENABLING WEAK LLMS TO JUDGE RESPONSE RELIABILITY VIA META RANKING

## ABSTRACT

Despite the strong performance of large language models (LLMs) across a wide range of tasks, they still have reliability issues. Previous studies indicate that strong LLMs like GPT-4-turbo excel in evaluating the reliability of responses from LLMs, but face efficiency and local deployment issues. Thus, to enable weak LLMs to effectively assess the reliability of LLM responses, we propose a novel cross-query-comparison-based method called *Meta Ranking* (MR). Unlike previous few-shot methods that solely based on in-context learning capabilities in LLMs, MR assesses reliability by pairwise ranking the target query-response pair with multiple reference query-response pairs. We found that MR is highly effective in error detection for LLM responses, that MR with weaker LLMs, which have lower task performance, results in higher judgement precision against baselines with the same or even stronger models. Moreover, the method requires as few as five reference samples and significantly improving efficiency. We further demonstrate that MR can enhance strong LLMs' performance in two practical applications: model cascading and instruction tuning. In model cascading, we combine open- and closed-source LLMs to achieve performance comparable to GPT-4-turbo with lower costs. In instruction tuning, we use MR for iterative training data filtering, significantly reducing data processing time and enabling LLaMA-7B and Phi-2 to surpass 13B models with fewer training tokens. These results underscore the high potential of MR in both efficiency and effectiveness.[1]

## 1 INTRODUCTION

Large language models (LLMs) have demonstrated strong performance in various tasks (OpenAI, 2023; Touvron et al., 2023b; Du et al., 2022). However, they still face reliability challenges. For example, these models often produce responses that seem plausible but are factually incorrect, a phenomenon known as "hallucination" (Huang et al., 2023). Fine-tuning and alignment techniques have been extensively studied to mitigate this issue (Ouyang et al., 2022; Wang et al., 2023; Rafailov et al., 2023; Yang et al., 2023; Gupta et al., 2024). Recent studies sadly demonstrate that hallucination is inevitable (Xu et al., 2024). Consequently, instead of resolving it directly, we focus on developing techniques to discriminate the reliability of responses from LLMs.

Recent research has highlighted the potential of strong LLMs in evaluating response reliability (Zheng et al., 2023a). Highly capable models, such as GPT-4 (OpenAI, 2023), have shown effective in assessing the quality of LLM responses through few-shot in-context learning (ICL) (Yin et al., 2023). However, these models are often prohibitively large, resulting in high computational and monetary costs. Also, most of these models are closed-source, which limits their deployment in local environments. On the other hand, weak models, are often better choices for efficiency and local setup. However, their performance is usually lower, probably due to the inherent low capacity in ICL (Figure 1 (a)). This raises a critical question: *Is it possible to enable weak LLMs to effectively judge the reliability of LLM responses despite the low task performance?*

To address this question, we propose a novel method named *Meta Ranking* (MR). Inspired from the idea of pairwise ranking on responses to the same query (Wang et al., 2024b; Ke et al., 2024; Zhu et al., 2023a), we raised a core hypothesis of MR, that the reliability of a response can be discerned

---

[1]The source code, data, and model checkpoints will be released.

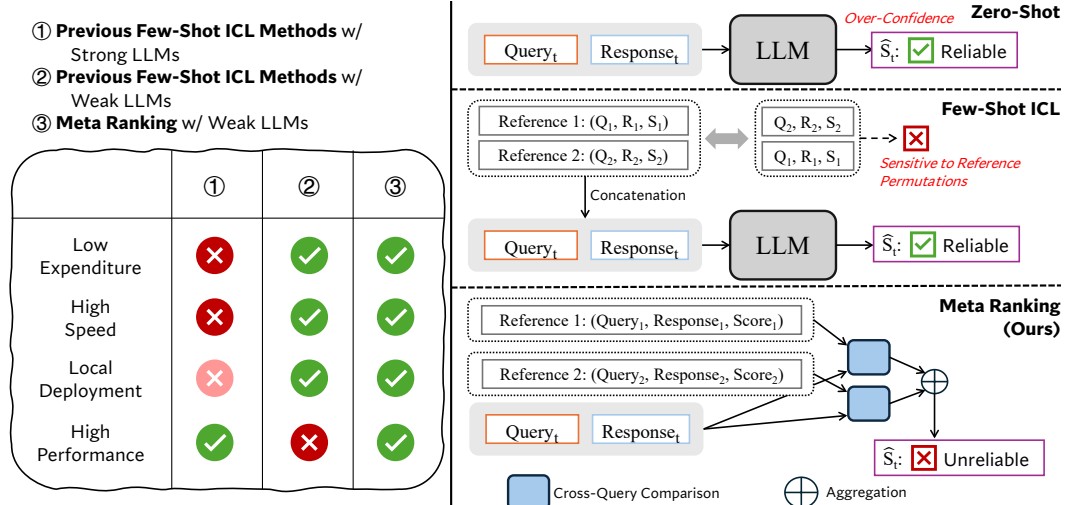

Figure 1: Overview of our proposed *Meta Ranking* (MR) method. (a) Left: The table summarizes MR and previous judgement methods with different backbone models. (b) Right: The sub-figure illustrates different methods. "$\widehat{S}_t$" denotes the estimated score for the target query-response pair. "Query$_i$" ($Q_i$), "Response$_i$" ($R_i$), and "Score$_i$" ($S_i$) ($i = 1, 2$) denote the references and its score (e.g., +1 for correct and -1 for incorrect responses). MR takes two query-response pairs for cross-query comparison on reliability with language models, then aggregates the estimated score of the target query and response.

by comparing the query-response pair with other pairs of known reliability. Unlike traditional methods that let an LLM directly judge the response to a query, MR involves cross-query comparison of the target query-response pair with multiple reference pairs (Figure 1 (b)). Specifically, MR utilizes a fixed set of query-response pairs with pre-determined reliability scores as reference. For any given target query-response pair, the LLM determines whether this pair is more reliable than each of the reference pairs. A voting mechanism is then employed to aggregate these comparisons and reach a final judgment. Here, **"reliable" encompasses attributes such as correctness and quality as required by the context**. Theoretically, it avoids item perturbation problems in few-shot ICL (Zhao et al., 2021) and over-confidence on the target response in judgement (Xiong et al., 2024) for LMs. Empirical results demonstrate that MR enables weaker LLMs to effectively judge LLM responses on reasoning tasks, resulting in better judgement precision scores with low task performance.

Moreover, we showcase the application of MR with a weak LLM in two practical scenarios for validation: (1) enhancing LLM inference through *model cascading* between open- and closed-source LLMs, where queries are routed to the appropriate LLM based on reliability assessments. It demands high efficiency of the judgement process. With MR, the model cascading achieves performance comparable to GPT-4-turbo while consuming less than half API tokens. And (2) iteratively filtering training datasets to improve *instruction tuning*, which prefers local deployment and also, the efficiency. With a light-weight judge model, MR leads to advancements over existing data selection methods on the Alpaca-52k dataset (Taori et al., 2023), in terms of effectiveness and efficiency.

In summary, our contributions are threefold:

1. We introduce *Meta Ranking* (MR), a novel method for assessing the reliability of LLM responses through cross-query comparison with reference query-response pairs.

2. We demonstrate that MR enables weak LLMs to judge the reliability of LLM responses, surpassing previous uncertainty estimation and prompting methods even with some strong LLMs, on effectiveness and efficiency.

3. Additionally, we explore two practical applications of MR, improving strong LLMs in both inference and training, respectively. These results underscore the considerable potential of our proposed method in both efficiency and effectiveness.

## 2 META RANKING

This section demonstrates how cross-query comparisons could reveal the reliability of the target query-response pair with limited reference examples from the same source LLM. The intuition is as follows: Taking correctness assessment as an example, *the target pair is likely to be correct when ranked closer to a correct reference pair and higher to an incorrect pair, and vice versa*, as shown in Figure 2. Below, we outline the specific steps and considerations accordingly.

Formally, suppose we have $N$ reference query-response pairs

$$\mathcal{X} = \{(Q_i, R_i, S_i)\}, \tag{1}$$

where $i = 1, \cdots, N$, $Q_i$ and $R_i$ are the $i$-th reference query and response, respectively. For each pair of $(Q_i, R_i)$, we have a score $S_i$ that represents its reliability. We aim to estimate the reliability $S_t$ of a target response $R_t$ to the target query $Q_t$, by an estimation score $\widehat{S}_t$. For binary classification scenarios (e.g., correctness assessment), $S_i, S_t \in \{+1, -1\}$, where $+1$ denotes the response is True and $-1$ denotes False.

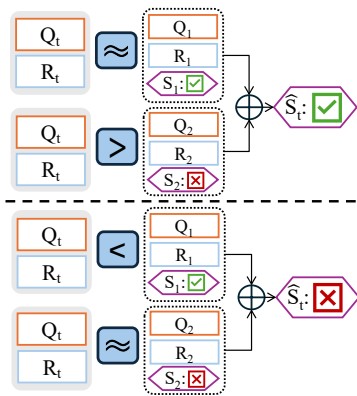

Figure 2: Example illustrations of MR process. The correctness of the target response ($R_t$) is considered according to comparisons with reference query-response pairs.

**Cross-Query Comparison** The basic operation of *Meta Ranking* is to compare the target query-response pair with each of the reference query-response pairs. For brevity, we denote the target query-response pair as $P_t = (Q_t, R_t)$, and the $i$-th reference query-response pair as $P_i = (Q_i, R_i)$. Then, the cross-query comparison operation and its result are denoted as follows:

$$r_i = \mathrm{MR}\left(P_t, P_i\right), i = 1, \cdots, N, \tag{2}$$

where $r_i \in \{+1, 0, -1\}$, $+1$, $0$, and $-1$ denote the target pair is better than, equal to, or worse than the $i$-th reference pair, respectively. In practice, $\mathrm{MR}(\cdot, \cdot)$ is implemented by directly prompting LLMs or using the relative magnitude of quality estimation scores of each response to its query.

**Aggregation** The final judgement is obtained by aggregating the comparison results to arrive at the estimated reliability score of the target query-response pair. Specifically, we will upvote if the target air is ranked higher than a reference pair, i.e., $r_i = +1$, and downvote when ranked lower. Also, ranking higher than a correct reference and than an incorrect reference will result in different voting values. Thus, the process is a kind of weighted voting. For each comparison between the target and the $i$-th reference pair, the individual voting value is

$$s_i = S_i \cdot \delta_{\mathrm{sgn}(S_i) \cdot r_i}, i = 1, \cdots, N, \tag{3}$$

where $\mathrm{sgn}(\cdot)$ is the sign function, $\mathrm{sgn}(S_i) \cdot r_i \in \{+1, 0, -1\}$, and $\delta_{+1}$, $\delta_0$, and $\delta_{-1}$ are hyperparameters. For instance, in terms of correctness, $\delta_{+1}$ is the absolute voting value when the target pair is ranked higher than a correct reference ($\mathrm{sgn}(S_i) = +1$, $r_i = +1$), or lower than an incorrect one ($\mathrm{sgn}(S_i) = -1$, $r_i = -1$). The rationale of the hyper-parameter as the voting weight is enumerated in Table 5. Note that we require that $\delta_{+1} > 0, \delta_{-1} < 0$. Formally, we denote the total vote value as $s$:

$$s = \sum_{i=1}^{N} s_i = \sum_{i=1}^{N} S_i \cdot \delta_{\mathrm{sgn}(S_i) \cdot r_i}, \tag{4}$$

And we say the target response is reliable if $s \geq 0$ and unreliable otherwise. Thus, the estimated target reliability score $\widehat{S}_t \approx \mathrm{sgn}(s)$ for correctness assessment. The entire algorithmic process is shown in Appendix C. In practice, $N$ is usually small due to efficiency and the limited labeled data.

For theoretical validation, when cross-query comparison reveals the actual relation between $S_i$ and $S_t$, we show that $\mathrm{sgn}(s) \approx \mathrm{sgn}(S_t - S_{\mathrm{avg}})$, where $S_{\mathrm{avg}} = \frac{\sum_i S_i}{N}$ under reasonable constraints in Appendix D.1. Hence, a negative $s$ means subpar reliability of the target response, and vice versa.

Under the formulation, there are several interesting properties of MR. First, **MR is model-agnostic and permutation-agnostic towards references**, which is different from few-shot ICL methods that fluctuate with the order of examples (Zhao et al., 2021). Second, **MR alleviates the over-confident**

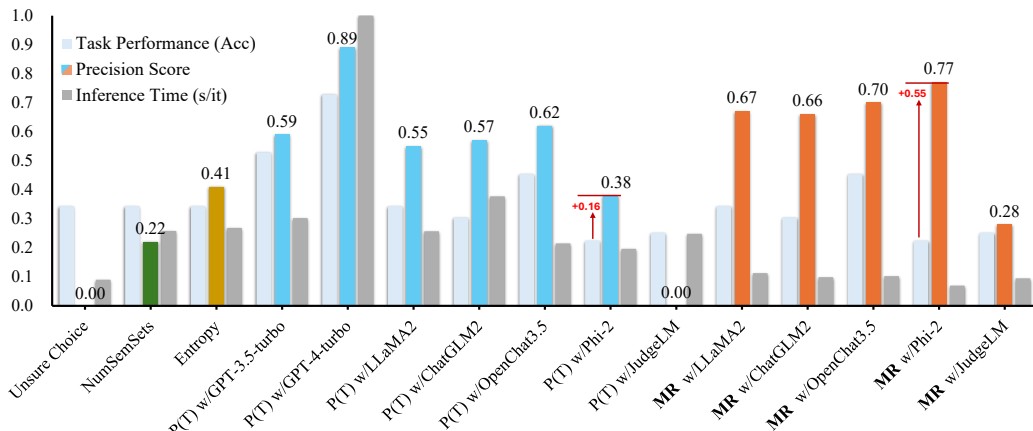

Figure 3: The task performance (light blue) in accuracy, judgement precision scores and inference time (gray) in error detection experiments for target responses from LLaMA-2 on the MMLU dataset. We used examples in the development set as reference for few-shot methods. We observed larger gap between task performance when using MR than baseline methods.

**issue in LLM judgement** (Xiong et al., 2024) that LLMs are too confident in their own responses. In MR, we provide two responses on different queries from the same LLM for the judge (equation 2). Thus, in theory, MR should not be over-confident on any side ($P_t$ or $P_i$ in equation 2), regardless that the responses are from the same model as the judge LLM or not. Still, judgement from LLMs directly might incorporate positional bias between the target and reference pair. We randomly swap them during prompt assembly to mitigate the issue. Finally, **MR could be extended to continuous metrics** ($S_i \in \mathbb{R}$, e.g., BLEU (Papineni et al., 2002)) **directly without modification**, and the final judgement of the response reliability is still determined by $\mathrm{sgn}(s)$.

## 3    MAIN EXPERIMENT: ERROR DETECTION WITH META RANKING

In the following section, we empirically demonstrate that *Meta Ranking* can effectively judge the reliability of LLM responses concerning **correctness**. We leverage error detection on responses from LLMs in reasoning tasks for validation. Our findings indicate that the MR approach achieves high judgement precision despite the low task performance, response accuracies, and languages.

### 3.1    SETTINGS

The error detection task requires identifying whether a response from an LLM is incorrect given the query. The settings are as follows, and implementation details are in Appendix C.1:

**Datasets**: To validate the effectiveness of methods on diverse tasks, we conduct experiments on 14 reasoning datasets, including subjects of "STEM", "Arithmetics", "Humanities", "Social Science", "Chinese Culture", etc. Specifically, we selected 8 test sets in MMLU (Hendrycks et al., 2021b;a), 5 test sets in CMMLU (Li et al., 2024a), and the test set from GSM8K (Cobbe et al., 2021). The MMLU and CMMLU datasets are multi-choice and the GSM8K dataset contains open-ended math queries. We select 5 few-shot samples for each dataset from its development or training set.

**Response Generation**: For different model accuracies, we chose `LLaMA-2-chat-7B` (Touvron et al., 2023b) and `OpenChat-3.5` (Wang et al., 2024a) to zero-shot generate responses for English queries (MMLU and GSM8K), and `ChatGLM-2-6B` (Zeng et al., 2023) and `Yi-6B-Chat` (01.AI, 2023) for Chinese ones (CMMLU). For each multi-choice query, we first generate the reasoning path for judgement, and then extract the choice by calling the generator model again.

**MR Settings**: We prompted LLaMA-2, ChatGLM-2, OpenChat-3.5, GPT-3.5-turbo, and Phi-2 to judge on different query-response pairs with *Meta Ranking*. We also tested an LLM-as-a-Judge-tuned model `JudgeLM-7B-v1` (Zhu et al., 2023a) in MR to see if the fine-tuning for the evaluation of responses to the same query helps. By setting each label $S_i$ with the value $\pm 1$, where $+1$ and $-1$ denote True and False, we apply MR on this task to identify incorrect responses.

Table 1: The micro precision scores on error detection experiments on the MMLU, CMMLU, and GSM8K datasets with responses generated by different LLMs. Models in the second row of the header are sources of responses and judge models are specified for each method. TP denotes the task performance of the judge model in accuracy. The number in the parentheses denotes the gap between the judgment precision and the task performance. The **bold font** denotes best results. *Due to low instruction following capacity of Phi-2, we barely extract valid answers from its zero-shot generation on GSM8K.*

| Method | MMLU (En) | | CMMLU (Zh) | | GSM8K (Math) | |
|---|---|---|---|---|---|---|
| | LLaMA-2 | OpenChat-3.5 | ChatGLM-2 | Yi | LLaMA-2 | OpenChat-3.5 |
| TP of Phi-2 | 0.22 | | 0.25 | | 0.00 | |
| TP of OpenChat-3.5 | 0.45 | | 0.40 | | 0.59 | |
| TP of GPT-3.5-turbo | 0.53 | | 0.54 | | 0.93 | |
| *Random Selection* | *0.50* | *0.50* | *0.50* | *0.50* | *0.50* | *0.50* |
| P(T) w/ OpenChat-3.5 | 0.62 (+0.17) | 0.38 (-0.07) | 0.31 (-0.09) | 0.35 (-0.05) | 0.39 (-0.20) | 0.11 (-0.48) |
| P(T) w/ GPT-3.5-turbo | 0.59 (+0.06) | 0.65 (+0.12) | 0.22 (-0.32) | 0.21 (-0.33) | 0.80 (**-0.13**) | 0.48 (-0.45) |
| **MR** w/ Phi-2 | 0.77 (**+0.55**) | 0.73 (**+0.51**) | 0.69 (**+0.44**) | 0.52 (**+0.27**) | **0.93** (**+0.93**) | **0.91** (**+0.91**) |
| **MR** w/ GPT-3.5-turbo | **0.78** (+0.25) | **0.79** (+0.26) | **0.75** (+0.21) | **0.73** (+0.19) | 0.64 (-0.29) | 0.87 (**-0.06**) |

**Baselines**: We compare our method against several baselines with few-shot ICL to ensure a comprehensive evaluation, including (1) appending an **Unsure Choice** (Kadavath et al., 2022), (2) a black-box uncertainty estimation method **NumSemSets** (Kuhn et al., 2023), (3) a white-box method **Entropy** (Han et al., 2024), and (4) P(True) (**P(T)**) (Kadavath et al., 2022) which directly asks an LLM about the correctness of a query-response pair. Additionally, we have discussed other uncertainty-based methods (Lin et al., 2024) in Appendix E.1.

**Evaluation Metrics**: We adopted micro scores calculated across MMLU, CMMLU, and GSM8K datasets. We report precision of judgement since it is essential to pick out more unreliable responses, and seconds per iteration for inference time on a single A800 GPU (Figure 3). Inference time are normalized with P(T) with GPT-4-turbo as the unit. And F1 scores are reported in Appendix C.1 to ensure the balance of judgement. AUROC-style metrics are not applicable because MR uses a static threshold and calibration methods (Han et al., 2024) determine the threshold of baseline methods. We also measured task performance of judge models in accuracy.

## 3.2 DISCUSSION

**Meta Ranking is Effective across Different LLM Backbones** The effectiveness of MR might attribute to the position agnosticism to reference and judgement without over-confidence. In Figure 3, we report results of all baselines and MR in error detection on LLaMA-2-generated responses, and the actual performance of LLMs on MMLU. Impressively, we found that MR with Phi-2 notably exceeds all baseline methods, except for P(T) with GPT-4-turbo, reaching a precision score of 0.77, more than double the performance of P(T) with Phi-2 and reaching 88% GPT-4-turbo performance. With LLaMA-2 and ChatGLM-2, MR exceeds P(T) significantly. However, JudgeLM performs not as well as other pretrained or general aligned LLMs in the MR results and fails to generalize for the P(T) method. It might be the fine-tuning process limits its generalization. Overall, MR consistently outperforms the random baseline to a greater margin than P(T), and has a greater gap between judgement precision and task performance. However, GPT-4-turbo detect the error most accurately even with P(T), probably relying on its strong reasoning capabilities and generalizability (OpenAI, 2023). We also depict F1 scores in Appendix C.1 to indicate MR is not biased to identify most responses as incorrect ones, which could result in 1.00 judgement precision but extremely low F1.

**Meta Ranking with Weak LLMs is Efficient and Robust** In Figure 3 and Table 1, the performance of P(T) across different models displays a positive correlation with their actual performance on reasoning tasks, while MR has demonstrated strong robustness and better efficiency across models with different capabilities. The improvement of efficiency is because the utilization of reference demonstration in MR is through cross-query comparisons, which do not introduce long inputs from in-context examples as P(T), saving the inference time from quadratic computational complexity of Transformer (Vaswani et al., 2017) with the input length. We further investigate the performance of Meta Ranking with LLMs with different capabilities across languages and response accuracies. In Table 1, we select top performing P(T) and MR methods and report the precision score across all target responses. We omit GPT-4-turbo for the cost and its unparalleled capabilities to open-source

models we have tested. From results, P(T) with OpenChat-3.5 performs worse than random selection when facing more accurate responses on MMLU. In contrast, MR shows significant robustness with weak models, e.g., the 2.7B Phi-2, impressively surpassing P(T) with GPT-3.5-turbo and OpenChat-3.5 on responses in different accuracy levels.

**Meta Ranking Demands Fewer Reference Pairs**
In Figure 4, we illustrate the change of precision scores with the number of reference pairs, where MR exhibits that it could function with far less labeled data compared to previous methods. Notably, other uncertainty-based methods are incompatible with the 1-shot setting since there are usually no correct examples for calibration. Upon the ablation study, we observe that reducing five examples to one leads to a slightly decreased performance of MR, indicating the robustness. Also, the result highlights the effectiveness of MR with limited labeled data compared to P(T). Without reference examples, P(T) faces a great performance drop, show-

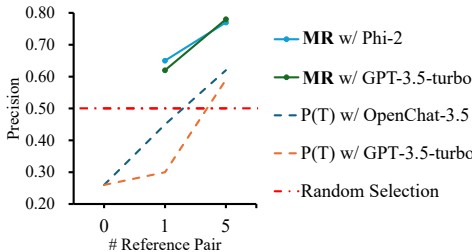

Figure 4: The change of precision scores with the number of reference pairs on the MMLU dataset with target responses from LLaMA-2.

ing inferior performance to random selection probably because of over-confidence. It is also worth noting that uncertainty-based methods like NumSemSets and Entropy usually require hundreds of examples to calibrate the score threshold (Han et al., 2024; Mielke et al., 2022), explaining the relative low results of these methods in Figure 3 when there are only five labeled samples.

**Meta Ranking Generalizes to Non-English and Open-ended Tasks** In Table 1, the overall results on Chinese reasoning problems are lower than on English ones, showing that non-English languages have negative impact. Despite this, MR exhibits strong robustness across languages, while P(T) performs worse than random selection in all CMMLU results. Results on Japanese (Appendix E.2) show a similar trend. Meanwhile, Table 1 also shows that MR performs well on open-ended arithmetic tasks, except MR /w Phi-2 demonstrating a bit of biased judgements. The F1 score of MR /w Phi-2 on GSM8K is 0.85 for responses from LLaMA-2, but drops to 0.55 for OpenChat-3.5, although it outperforms most baselines (Appendix C.1).

## 4 APPLICATIONS OF META RANKING

In this section, we present two practical applications to further validate the effectiveness of *Meta Ranking* **beyond mere correctness**, as shown in Figure 5. Each application is implemented by collecting **reference query-response pairs** and setting **reference reliability scores**. (a) With the assessment of the reliability of responses from open-source LLMs to the given queries, we identify and route unsolved queries to stronger closed-source LLMs. It could achieve better efficiency and remaining performance of closed-source models. However, it demands the model used for judgement is weaker than the open-source LLM in deployment, otherwise it is better in place to respond to queries. (b) By evaluating the quality of instruction data, we can refine the supervised fine-tuning (SFT) for LLM instruction tuning, whose key factor is the quality of training data (Liu et al., 2024). By filtering low-quality data after each epoch and further introducing post-SFT training with mere instruction data, we achieve significant improvement against state-of-the-art SFT data selection methods. For this data-related application, the method is better to be locally deployed and with high efficiency. Thus, we decide MR with weak LLMs is suitable for judgement for these.

### 4.1 MODEL CASCADING

Since LLMs exhibit varying performance across various tasks, we propose using MR within a model cascading system. As depicted in Figure 5 (a), this system employs MR to assess the reliability of generated responses from open-source LLMs. Queries with responses deemed unreliable by MR are routed to more powerful, but also more costly, closed-source LLMs for better answers. This system aims to achieve performance similar to closed-source models with improved efficiency.

#### 4.1.1 IMPLEMENTATION

Assuming both the development and the test sets are drawn from the same underlying distribution. Given that the MR method requires reference query-response pairs, we first feed the queries in the

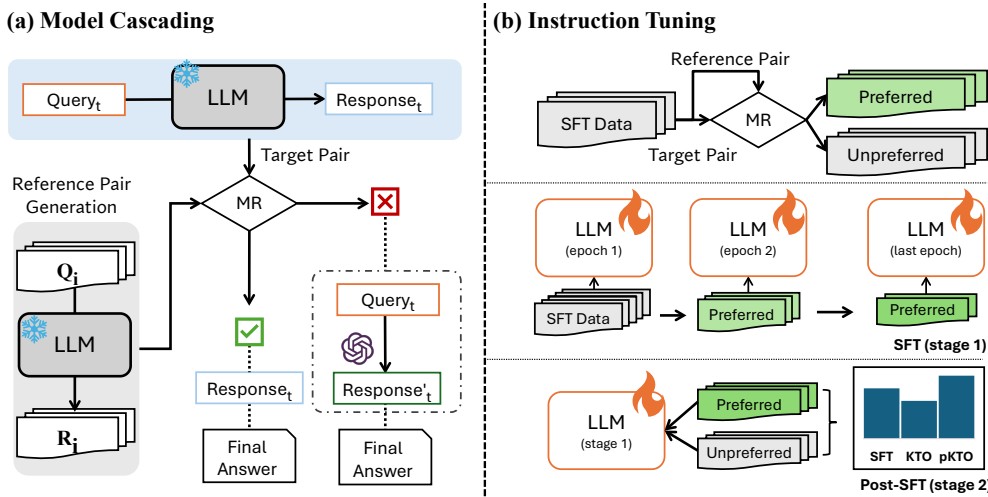

Figure 5: Two applications of *Meta Ranking* for inference- and training-time LLM enhancement, respectively. (a) Model Cascading (left): MR identifies reliability of responses and routes unsolved queries from open-source LLMs towards closed-source LLMs for better results (Response$_t$ → Response$'_t$). (b) Instruction Tuning (right): MR filters low-quality data after each epoch in SFT and then tune LLMs with low- and high-quality data. MR results depend on the reference pairs generated from the LLM on samples of the training dataset. $Q_i$, $R_i$ denote reference query-response pairs for the MR algorithm, and Query$_t$, Response$_t$ denote the target pair.

development set to the open-source LLM and evaluate the generated responses against the ground truth. Formally, for every query $Q_i$, let $R_i^{(\theta)}$ represent the response generated by an open-source LLM (parameterized by $\theta$), and $S_i^{(\theta)}$ denote the evaluation result according to the ground truth to $Q_i$ and an appropriate metric. By applying the model to each query in the development set with $N$ samples, we produce responses $\left\{R_i^{(\theta)}\right\}_{i=1}^N$ and form a set of **reference query-response pairs** $\mathcal{X} = \left\{P_i^{(\theta)} = (Q_i, R_i^{(\theta)})\right\}_{i=1}^N$, along with associated evaluation results $\left\{S_i^{(\theta)}\right\}_{i=1}^N$.

There are **two reasonable ways to derive reliability scores** for $\mathcal{X}$ in MR. The first is to directly define $S_i \triangleq S_i^{(\theta)}$, which we term MR($\theta$). The second option is to compute the responses from the closed-source LLM ($\Theta$) and their evaluation results $\left\{S_i^{(\Theta)}\right\}_{i=1}^N$, and define the score as follows:

$$S_i \triangleq S_i^{(\theta)} - S_i^{(\Theta)}, \tag{5}$$

which denotes the extent that the response from open-source LLMs is better than the one from closed-source LLM. Since a query will only be routed when the predicted reliability score is negative, this definition matches the principle that the model cascading only causes performance improvement when closed-source LLM performs better. We denote it by MR($\Delta$), which considers the gap between open- and closed-source LLMs.

Accordingly, we can obtain estimated reliability for $P_t^{(\theta)}$ from the MR approach during inference on test sets. If the assessment indicates $R_t$ as an unreliable response, we direct $Q_t$ to a closed-source large language model (such as GPT-4-turbo) to secure a more precise response. Conversely, we preserve the original response for a positive MR result. Ideally, this approach can enhance performance with moderate costs, as it is generally observed that a poorly accurate response from open-source LLMs often corresponds to a difficult query, which requires strong LLMs to respond.

### 4.1.2 EXPERIMENT

**Settings** We leverage reasoning and translation tasks to validate the effectiveness of model cascading. We use the same datasets for reasoning tasks as Section 3 and randomly sampled test set and

Table 2: The overall results in model cascading experiments. MR is implemented with Phi-2, whose results are gray. "Average" denotes the macro average value across tasks. The **bold font** denotes the best result using model cascading and the underlined numbers denote the best result for each task. The number in the parentheses denotes the improvement over the best among open-source LLMs and the ensemble baseline without model cascading. For notation, serial numbers represent LLMs, e.g., "①/② + ③" represents model cascading from LLaMA-2 or ChatGLM-2 to GPT-3.5-turbo.

| Model | Routing Strategy | Reasoning | | Translation | | Average | #Token (API) (Relative Value) |
| | | English | Chinese | Zh-En | En-Zh | | |
| --- | --- | --- | --- | --- | --- | --- | --- |
| *Phi-2* | *-* | *22.48* | *24.84* | *42.32* | *23.18* | *28.21* | *-* |
| LLaMA-2 (①) | | 34.37 | 32.86 | 58.53 | 51.33 | 44.27 | |
| ChatGLM-2 (②) | - | 30.48 | 49.71 | 58.43 | 63.30 | 50.48 | - |
| Ensemble (①&②) | | 35.44 | 43.47 | 31.35 | 46.54 | 39.20 | |
| GPT-3.5-turbo (③) | | 52.91 | 54.09 | 63.67 | 69.14 | 59.95 | 1.00 |
| ①/② + ③ | Entropy | 38.75 | 47.03 | 58.32 | 65.45 | 52.39 (+1.91) | 0.38 |
| | Random | 41.89 | 48.32 | 58.97 | 64.83 | 53.50 (+3.02) | 0.45 |
| | **MR($\Delta$)** | 44.30 | 48.67 | 59.80 | 65.87 | 54.66 (+4.18) | **0.24** |
| | **MR($\theta$)** | **48.62** | **52.76** | **61.36** | **67.10** | **57.46** (+6.98) | 0.42 |
| OpenChat-3.5 (④) | | 45.42 | 40.19 | 61.35 | 60.77 | 51.93 | |
| Yi (⑤) | - | 42.08 | 61.96 | 60.87 | 62.07 | 56.74 | - |
| Ensemble (④&⑤) | | 45.68 | 48.24 | 11.16 | 62.90 | 41.99 | |
| GPT-4-turbo (⑥) | | 72.86 | 62.82 | 64.73 | 69.95 | 67.59 | 1.00 |
| ④/⑤ + ⑥ | Random | 46.64 | **61.96** | 61.29 | 63.74 | 58.41 (+1.67) | 0.44 |
| | **MR($\Delta$)** | 57.68 | 61.93 | 61.60 | 67.61 | 62.21 (+5.47) | **0.23** |
| | **MR($\theta$)** | **64.68** | 61.93 | **62.60** | **68.11** | **64.33** (+7.59) | 0.43 |

few-shot samples from the FLORES-200 dataset (Guzmán et al., 2019) for translation tasks. To implement cross-query comparisons in MR, we prompt Phi-2 in reasoning tasks, and in translation tasks, we adopt `wmt22-cometkiwi-da` (Rei et al., 2022b) for reference-free quality estimation and thus compare the estimated scores between translations. Due to the diverse LLM capabilities and language biases, we have tested two combinations: (1) LLaMA-2 for English and ChatGLM-2 for Chinese tasks, with GPT-3.5-turbo as the closed-source model; (2) OpenChat-3.5 for English and Yi for Chinese problems, with GPT-4-turbo as the closed-source model. For baselines, we validate the logits ensemble of the open-source models and implement model cascading with strategies of entropy-based uncertainty estimation (Han et al., 2024). We omit the result of the latter for OpenChat-3.5 and Yi since it results in almost no routed queries, probably because the calibrated uncertainty threshold is too high to determine false answers.

**Effectiveness of Cascading Guidance from MR** In Table 2, we report the average accuracy on reasoning tasks and the average among BLEU, BLEURT (Sellam et al., 2020; Pu et al., 2021), and COMET (Rei et al., 2022a) scores on translation tasks. With the model cascading approach, we observe significant improvement against single open-source LLMs. MR($\theta$) and MR($\Delta$) manage to gain the highest performance improvement across tasks and languages with less than half token consumption. Moreover, MR($\Delta$) consistently outperforms random selection with nearly half of the token consumption, demonstrating the effectiveness of equation 5. We also demonstrate actual deployment costs and inference speeds in Appendix C.2 and found MR-based model cascading costs much lower in monetary expenditure.

**Relation with Error Detection Performance** The performance of our model cascading mechanism is closely related to the effectiveness of error detection. For instance, in Table 2, MR outperforms Random and Entropy on absolute performance and token consumption, which denotes MR detects errors in a larger quantity and with higher precision. It suggests that error detection with MR is also robust on open-ended generation tasks with continuous metrics, e.g., translation. For tasks whose responses could be more effectively validated through rule-based checker, e.g., code translation, we also validate the performance of MR in Appendix E.4, and the whole system even outperforms the closed-source model with lower token costs.

## 4.2 INSTRUCTION TUNING

Recent studies show that the quality of instruction data is essential to SFT performance (Zhou et al., 2023). For better instruction tuning on LLMs, we introduce an iterative training data filtering process

Table 3: Results on instruction tuning experiments, where MR is implemented with Phi-2. The **bold font** denotes best results. "Full" denotes the whole dataset.

| Method | MT-Bench | AlpacaEval 2.0 | #Token (M) |
|---|---|---|---|
| Alpaca-13B | 4.53 | 2.65 | - |
| Phi-2 | 4.52 | 2.34 | - |
| LLaMA-7B | 2.62 | 0.43 | - |
| *Phi-2-Based Results* | | | |
| Full | 4.42 | 3.26 | 13.293 |
| Longest | 4.56 | 3.32 | 1.008 |
| Deita | 4.33 | 3.18 | 9.609 |
| Deita (9k) | 4.64 | 3.29 | 3.981 |
| **MR** (Stage 1) | 4.70 | **3.56** | 7.509 |
| **+ pKTO** (Stage 2) | **4.77** | 3.47 | 1.205 |
| *LLaMA-7B-Based Results* | | | |
| Full | 4.36 | 2.53 | 13.293 |
| Longest | 4.18 | 2.35 | 1.008 |
| Deita | 4.37 | 2.60 | 9.609 |
| Deita (9k) | 4.48 | 2.86 | 3.981 |
| **MR** (Stage 1) | 4.52 | 2.93 | 2.412 |
| **+ pKTO** (Stage 2) | **4.69** | **3.24** | 0.907 |

Table 4: The multi-turn evaluation results from MT-Bench on instruction tuning experiments, which is unfolded from the second column in Table 3. "Avg." denotes the average score of different turns in MT-Bench. The **bold font** denotes the best result for each base model.

| Method | MT-Bench | | |
|---|---|---|---|
| | Turn 1 | Turn 2 | Avg. |
| Alpaca-13B | 4.98 | 4.09 | 4.53 |
| Phi-2 | 6.37 | 2.66 | 4.52 |
| LLaMA-7B | 3.30 | 1.94 | 2.62 |
| *Phi-2-Based post-SFT Results* | | | |
| **MR** (Stage 1) | **6.33** | 3.08 | 4.70 |
| + KTO (Stage 2) | 6.12 | 2.94 | 4.53 |
| + pKTO (Stage 2) | 6.27 | **3.26** | **4.77** |
| *LLaMA-7B-Based post-SFT Results* | | | |
| **MR** (Stage 1) | **5.47** | 3.58 | 4.52 |
| + KTO (Stage 2) | 5.15 | 3.60 | 4.38 |
| + pKTO (Stage 2) | 5.45 | **3.93** | **4.69** |

based on *Meta Ranking* and a post-SFT training stage, as shown in Figure 5 (b). The basic intuition is to continuously filter low-quality data, letting LLMs concisely learn from more reliable and fewer training samples at the first stage, and utilize less reliable data samples at the second stage for post-SFT contrastive learning. MR makes it possible by judging the quality of instruction data rapidly with generated responses from an LLM that reflect its capabilities during training, and is better for local deployment with weak LLMs.

### 4.2.1 IMPLEMENTATION

The application contains two stages (Figure 5 (b)): (1) SFT with MR guided data selection and (2) post-SFT training with both estimated low- and high-quality data from the last epoch at stage 1.

For the first stage, besides regular SFT, we extract a small set of queries from the training set and, after each epoch, ask the tuned LLM to respond to those queries. With **the generated responses and the queries as reference**, the MR method could judge whether each sample in the original training dataset matches the quality of the reference. For simplicity in MR, we set **the reliability score of all reference pairs** to 1. Thus, we could filter training data samples that fail the judgment, i.e. unreliable, improving training efficiency and, potentially, LLM performance.

For the second stage, we want to utilize both the filtered low-quality and the high-quality data to further train the LLM. Since post-SFT training methods (e.g., PPO (Ouyang et al., 2022), DPO (Rafailov et al., 2023)) require multiple responses of diverse human preferences or quality to the same query, they are not compatible. Recently, Ethayarajh et al. (2024) proposed Kahneman-Tversky Optimization (KTO) to align LLMs towards desired and away from undesired query-response pairs contrastively. However, we are aware that their objective is misaligned with our requirement because the low-quality data is derived from the SFT dataset, which is not completely negative. Therefore, in order to incorporate both high- and low-quality data as positive and partially positive samples, we propose positive-KTO (pKTO). Intuitively, pKTO differs from KTO only in dealing with low-quality data, where pKTO regulates the reward of these data with MSE loss instead of decreasing it unlimitedly. Specifically, we use equal amount of low-quality and high-quality data samples from the last epoch in the first stage to fine-tune the LLM with the pKTO objective. Please refer to Appendix D.2 for the detailed implementation and comparisons with DPO and KTO.

### 4.2.2 EXPERIMENT

**Settings** In all experiments, we only use the Alpaca-52k (Taori et al., 2023) dataset with 52,002 samples from `text-davinci-003` (Brown et al., 2020), which is also the target pairs for MR. We utilize AlpacaEval 2.0 (Li et al., 2023) and MT-Bench (Zheng et al., 2023a) to benchmark instruction

following capabilities. We select strong baselines on SFT data selection, including Deita (Liu et al., 2024) and Longest (Zhao et al., 2024). Please refer to Appendix C.3 for implementation details. For base models, we choose `Phi-2` and `LLaMA-7B` (Touvron et al., 2023a) for instruction tuning. For Phi-2, our method starts with the original 52k dataset. For LLaMA, Chen et al. (2024) empirically find a 9k subset of Alpaca is the most proper for SFT. We thus adopt the scorer from Deita and extract the top 9k data, noted by Deita (9k), as the initial training set for LLaMA at stage 1.

**High-Quality SFT Data Filtering Guided by MR (Stage 1)** We report overall results and training tokens (calculated by LLaMA tokenizer) in Table 3. For each base model, we found MR guided iterative training data filtering leads to significant improvement in both benchmarks with fewer training tokens. In the first stage of MR guided data selection, our method iteratively filters low-quality data after each epoch. After stage 1, Phi-2 and LLaMA already surpass all baselines, indicating MR effectively picks high-quality training samples in a curriculum way (Bengio et al., 2009), which helps align LLMs better compared to selecting data at the beginning for SFT in baseline methods. We also observed significant lower data processing time of MR compared to Deita in Appendix C.3.

**Post-SFT Training through MR-Filtered Data (Stage 2)** We also notice a significant enhancement of stage 2 in Table 3, and report the detailed scores on each turn of LLM responses from MT-Bench in Table 4. From empirical results, pKTO enhances the second-turn communication of LLMs to a great extent while preserving the instruction-following abilities from SFT at stage 1, which is indicated by the slight drops in first-turn scores. In contrast, KTO fails at this setting, which is aligned with our hypothesis in Section 4.2.1. By incorporating the low- and high-quality data distinguished from MR, stage 2 further elicits LLMs' capacity, especially for multi-turn scenarios.

## 5 RELATED WORK

**Evaluation of LLM Responses** Extensive research has been conducted to evaluate responses from LLMs. Studies have focused on estimating uncertainty to gauge the potential reliability of LLM responses (Kuhn et al., 2023; Rafailov et al., 2023). Furthermore, LLMs are capable of providing uncertainty scores from itself by fine-tuning (Chen et al., 2023; Gupta et al., 2024), which usually requires an amount of training data, or black-box measurements (Lin et al., 2024). However, these methods often require an amount of labeled data for calibration to determine a threshold (Han et al., 2024). Additionally, the LLM-as-a-judge approach effectively assesses the accuracy of LLM responses from strong LLMs (Zheng et al., 2023a) with manual prompting rules. Contrarily, our *Meta Ranking* method leverages weak LLMs and a training-free judgment based on cross-query comparisons with much fewer examples.

**Model Cascading with LLMs** Recent studies on model cascading focus on how LLMs can selectively call tools or stronger models only in difficult situations for better efficiency. Tool calls or another trial happens on external feedback from environment (Lin et al., 2023; Shinn et al., 2023). For tasks with explicit criteria, e.g., coding, LLMs can call stronger models after their failure (Zhang et al., 2023; Yue et al., 2024). Selection can also be achieved through fine-tuning (Erbacher et al., 2024) or uncertainty estimation (Han et al., 2024; Gupta et al., 2024). In our approach, MR route queries on complicated open-ended tasks, and empirical results are demonstrated in Section 4.1.2.

**Data-Efficient Training for LLMs** Coresets (Mirzasoleiman et al., 2020) are used in machine learning for a long period. For LLMs, several data selection methods are developed for SFT (Liu et al., 2024; Zhou et al., 2023; Li et al., 2024b; Chen et al., 2024) and post-SFT stages (Gulcehre et al., 2023; Aksitov et al., 2024). Inspired by the latter, we introduce iteratively filtering SFT data based on MR results after each epoch, and used in post-SFT training.

## 6 CONCLUSION

We present *Meta Ranking* (MR), a novel method effectively enabling weak LLMs to judge the reliability of LLM responses. By comparing a target query-response pair with a small number of reference pairs, MR outperforms strong baselines in error detection without fine-tuning. Furthermore, the method significantly enhances strong LLMs' performance and efficiency in two practical application scenarios, model cascading and instruction tuning. These findings highlight the potential of MR for broader inference- and training-time applications with LLMs.

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

## A  LIMITATIONS

There are several limitations to our work that we would like to acknowledge:

First, we have not explored deep in the compatibility between the model training process and the *Meta Ranking* (MR) method. Different model training strategies may affect the effectiveness of MR. It is an interesting direction for future work to study how to better integrate MR with the alignment process (e.g., SFT and post-SFT training) on LLMs.

Also, we have not focused on finding potential applications of the *Meta Ranking* method for strong models. Our current experiments focus on enabling weak LLMs to judge the reliability of LLM responses due to its superior efficiency and effectiveness. It remains an open question of what practical usage could incorporate MR with strong models like GPT-4-turbo. Exploring the potential applications of MR for strong models is also a direction for future work.

Last but not least, inaccurate judgments in sensitive domains like healthcare could lead to erroneous advice or misdiagnosis, endangering patient's health. In finance, inaccurate judgments could result in flawed financial advice or risk assessments, leading to substantial economic losses.

In conclusion, while our proposed *Meta Ranking* method has shown promising results in enabling weak LLMs to judge the reliability of LLM responses and enhancing LLM performance in practical applications, there are still spaces to be explored. We hope that future research can address these limitations and further improve the method.

## B  BROADER IMPACT

The *Meta Ranking* (MR) method presented in this paper has the potential to significantly influence the field of LLMs and their applications. Here, we discuss the broader impact of our work in several key areas:

**Data and LLM Response Curation:** Meta Ranking enables the use of smaller, less resource-intensive models for response judgement, which previously required large, expensive models. Thus, MR is more cost-efficient and practical for landing in real-world scenarios. Also, MR does not inherently incorporate risks in the methodology, while the language model MR utilizes could contain potential risks from its pre-train data. The application of MR, including model cascading and instruction tuning, may result in risky results due to the nature of the application. However, MR could actually be used in risk mitigation for LLMs by identifying and filtering LLM responses with potential risks.

**LLM Inference and Training:** The MR method can improve the efficiency and effectiveness of LLM inference and training. By routing queries to the most appropriate LLMs based on reliability assessments, MR can save computational resources and improve response times, making LLMs more practical for real-world applications. Additionally, the iterative training data refinement enabled by MR can lead to more accurate and reliable LLMs, which is crucial for maintaining public trust in AI systems.

In conclusion, the Meta Ranking method not only enhances the capabilities of weak LLMs but also has the potential to transform how we develop, deploy, and interact with AI systems, leading to a more reliable, efficient, and equitable integration of AI in various aspects of our lives.

## C  IMPLEMENTATION DETAILS

---

**Algorithm 1:** Meta Ranking

**Input**  : Target query-response pair $P_\text{t} = (Q_\text{t}, R_\text{t})$, reference query-response pairs $\mathcal{X} = \{P_i = (Q_i, R_i)\}_{i=1}^N$, the reliability score $S_i$ for each $P_i$, and hyper-parameters $\delta_{+1}$, $\delta_0$, and $\delta_{-1}$

**Output:** A boolean indicator $\mathcal{I}$ of the reliability of the target response (True indicates reliable)

$s \leftarrow 0$;

**for** $i \leftarrow 1$ **to** $N$ **do**
  $r \leftarrow \text{MR}\,(P_\text{t}, P_i)$;
  $r \leftarrow \text{sgn}(S_i) \times r$;
  $s \leftarrow s + S_i \times \delta_r$;
**end**

**if** $s \geq 0$ **then** $\mathcal{I} \leftarrow$ True **else** $\mathcal{I} \leftarrow$ False;

---

We demonstrate the detailed process of *Meta Ranking* in Algorithm 1. And we list all rationales for the hyper-parameter $\delta$ as the absolute voting weight used in equation 3 under different conditions of $S_i, r_i$ in Table 5. Under this core technique, we elaborate on the implementation details of the experiments below.

For general settings, all experiments in this paper were conducted on two types of servers: 8*A800 and 8*V100. The A800 server is equipped with 8 NVIDIA `A800-SXM4-80GB` GPUs. The V100 server features 8 NVIDIA `Tesla V100-PCIE-32GB` GPUs. For the name of LLMs, we use Phi-2 to denote `Phi-2` (2.7B)[2] (Li et al., 2023), LLaMA to denote `LLaMA-7B`[3] (Touvron et al., 2023a), LLaMA-2 to denote `LLaMA-2-7B-chat`[4] (Touvron et al., 2023b), ChatGLM-2 for `ChatGLM2-6B`[5] (Zeng et al., 2023), OpenChat-3.5 for `OpenChat-3.5` (7B)[6] (Wang

---

[2] https://huggingface.co/microsoft/phi-2
[3] https://huggingface.co/huggyllama/llama-7b
[4] https://huggingface.co/meta-llama/Llama-2-7b-chat
[5] https://huggingface.co/THUDM/chatglm2-6b
[6] https://huggingface.co/openchat/openchat_3.5

Table 5: The rationales for the hyper-parameter $\delta$ as the absolute voting weight used in equation 3 under different conditions of $S_i, r_i$ in Algorithm 1.

| Hyper-Parameter | Value of $\mathrm{sgn}(S_i)$ | Value of $r_i$ | Rationale |
|---|---|---|---|
| $\delta_{+1}$ | +1 | +1 | The target query-response pair is better ($r_i = +1$) than a reliable reference ($\mathrm{sgn}(S_i) = +1$). |
| | -1 | -1 | The target query-response pair is worse ($r_i = -1$) than an unreliable reference ($\mathrm{sgn}(S_i) = -1$). |
| $\delta_0$ | +1 | 0 | The target query-response pair is equal ($r_i = 0$) to a reliable reference ($\mathrm{sgn}(S_i) = +1$). |
| | 0 | - | The target query-response pair is compared to a reference of unknown quality ($\mathrm{sgn}(S_i) = 0$). |
| | -1 | 0 | The target query-response pair is equal ($r_i = 0$) to an unreliable reference ($\mathrm{sgn}(S_i) = -1$). |
| $\delta_{-1}$ | +1 | -1 | The target query-response pair is worse ($r_i = -1$) than a reliable reference ($\mathrm{sgn}(S_i) = +1$). |
| | -1 | +1 | The target query-response pair is better ($r_i = +1$) than an unreliable reference ($\mathrm{sgn}(S_i) = -1$). |

et al., 2024a), Yi for `Yi-6B-Chat`[7] (01.AI, 2023), and GPT-3.5-turbo and GPT-4-turbo for `GPT-3.5-turbo-1106` and `GPT-4-1106-preview` (OpenAI, 2023). All open-source models are deployed with vLLM inference framework (Kwon et al., 2023) in fp16 precision. For reproduction, we set the `temperature` to 0 for LLM generation without specification. And the maximum number of tokens for LLM response generation is set to 512.

For clarification, we have ensured that the use of pretrained and instruction-tuned LLMs and datasets is consistent with their intended use and licenses. Furthermore, the derivatives of these data and instruction-tuned models based on MR should be used in consideration of the original access conditions and ethical guidelines.

## C.1 ERROR DETECTION

**Detailed Settings** For the MMLU dataset (Hendrycks et al., 2021b), we randomly selected subjects in each category, including "Abstract Algebra" and "College Mathematics" for STEM, "Prehistory" and "Moral Scenarios" for humanities, "Econometrics" and "Professional Psychology" for social sciences, and "Global Facts" and "Professional Accounting" for others. For the CMMLU dataset (Li et al., 2024a), we select "College Actuarial Science" for STEM, "World History" for humanities, "Security Study" for social sciences, "Traditional Chinese Medicine" for China-specific subjects, and "Human Sexuality" for others. For MMLU and CMMLU, we use all the five examples in the development set as the reference pairs, along with generated responses for the reference query-response pair in MR. For GSM8K (Cobbe et al., 2021), we randomly sampled five examples in the training set as reference. We use the original answer extractor to check whether the answer is correct from generator LLMs. We performed grid search for hyper-parameters $\delta_{+1}, \delta_0, \delta_{-1} \in \{\pm 1, \pm 0.5, \pm 0.25, 0\}$, and assign the hyper-parameter in MR as following: $\delta_{+1} = 1, \delta_0 = 1, \delta_{-1} = -0.5$ for MMLU, and $\delta_{+1} = 1, \delta_0 = 0.5, \delta_{-1} = -0.25$ for CMMLU and GSM8K. We calculate the accuracy by exactly matching the generated choice (e.g., A, B, C, or D) with the ground truth. We measured the inference speed on a single A800 GPU with a single worker.

**Baseline Implementations** For (1) **Unsure Choice** (Kadavath et al., 2022), we include an additional option, "`(E) Not Sure`", allowing the LLM to admit uncertainty in a zero-shot manner on questions it might answer incorrectly. For uncertainty measurement (2) **NumSemSets** (Kuhn et al., 2023), given that the responses of different choices inherently form semantic sets, we sample five times on each question with the same LLM with the `temperature` of 0.8, and decide on an incorrect answer if the number of semantic sets is larger than all correct examples, following the calibration method in Han et al. (2024). (3) **Entropy** (Han et al., 2024) measures the confi-

---

[7] https://huggingface.co/01-ai/Yi-6B-Chat

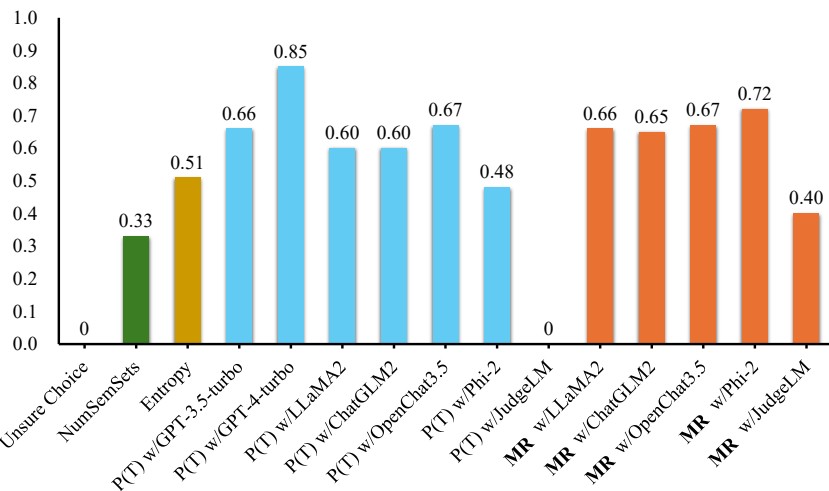

Figure 6: The F1 score of all methods on error detection experiments for target responses from LLaMA-2 on the MMLU dataset. The metric shows similar trends to the precision score in Figure 3.

dence of LLM responses in a white-box way, and picks out incorrect responses in the exact way as NumSemSets, that a response is considered incorrect when its uncertainty value is lower than all correct examples. Lastly, (4) **P(T)**: the P(True) (Kadavath et al., 2022) baseline directly asks an LLM about the correctness of a query-response pair, thus able to be implemented on different LLMs, including the aforementioned open-source LLMs and closed-source `GPT-3.5-turbo-1106` and `GPT-4-1106-preview`. We implement NumSemSets, Entropy, and P(T) in a few-shot manner, with the same five examples as MR. We shuffled the few-shot examples to eliminate the positional bias. For the one-shot experiments in Figure 4, we use the first sample.

**Biases in Error Detection**   To validate the potential biases in the error detection experiments (e.g., judging all responses as false ones) in Figure 3, we report micro F1 scores of the MMLU dataset in Figure 6 and F1 scors of the GSM8K in Table 6. We can observe that the F1 results approximately follow the trend of precision scores in Figure 3, and for MR methods, the F1 scores are close to or surpass GPT-3.5-turbo, demonstrating unbiased judgement for correct and incorrect responses.

Table 6: The F1 score on error detection experiments on the GSM8K dataset. Models in the second row of the header are sources of responses.

| Method | GSM8K | |
|---|---|---|
| | LLaMA-2 | OpenChat-3.5 |
| P(T) w/ OpenChat-3.5 | 0.56 | 0.19 |
| P(T) w/ GPT-3.5-turbo | 0.86 | 0.48 |
| **MR** w/ Phi-2 | 0.85 | 0.55 |
| **MR** w/ GPT-3.5-turbo | 0.77 | 0.57 |

## C.2   MODEL CASCADING

**Detailed Settings**   For the translation dataset construction, we randomly extracted 400 parallel sentences from the dev-test set as the test set and 20 sentences from the development set as few-shot samples, respectively, in Chinese and English from Flores-200 (Guzmán et al., 2019). For evaluation, we adopt SacreBLEU (Post, 2018) for BLEU calculation[8], BLEURT-20 (Pu et al., 2021) for BLEURT scores (Yan et al., 2023), and `wmt22-comet-da`[9] for COMET scores (Rei et al., 2022a). We report the detailed results of translation tasks in Table 7. For MR implementation, we use Phi-2 as the backbone model. We follow the same implementation of MR from error detection experiments (Appendix C.1) on reasoning tasks. For translation tasks, we set $\delta_{+1} = 1, \delta_0 = 0, \delta_{-1} = -1$ in Algorithm 1. We implement the cross-query comparison with the reference-free quality estimation model `wmt22-cometkiwi-da`[10] (Rei et al., 2022b).

---

[8]The signature is "nrefs:1+case:mixed+eff:no+smooth:exp +version:2.3.1".

[9]https://huggingface.co/Unbabel/wmt22-comet-da

[10]https://huggingface.co/Unbabel/wmt22-cometkiwi-da

Table 7: Detailed results on translation tasks in Table 2. "**#Token** (Local)" denotes the total number of prompt and generated tokens during inference of open-source LLMs.

| Model | Zh-En | | | En-Zh | | | #Token (Local) | #Token (API) |
|---|---|---|---|---|---|---|---|---|
| | BLEU | BLEURT | COMET | BLEU | BLEURT | COMET | $(\times 10^4)$ | $(\times 10^4)$ |
| Phi-2 | 8.1 | 49.60 | 69.27 | 2.5 | 23.28 | 43.77 | 49.82 | |
| LLaMA-2 (①) | 20.1 | 71.13 | 84.35 | 22.7 | 55.30 | 75.99 | 17.52 | |
| ChatGLM-2 (②) | 20.8 | 70.52 | 83.97 | 36.3 | 68.21 | 85.40 | 7.43 | - |
| OpenChat-3.5 (④) | 24.4 | 73.85 | 85.80 | 32.1 | 66.24 | 83.97 | 8.92 | |
| Yi (⑤) | 23.8 | 73.39 | 85.41 | 31.1 | 69.16 | 85.94 | 6.78 | |
| GPT-3.5-turbo (③) | 27.8 | 76.04 | 87.16 | 45.7 | 73.12 | 88.59 | | 8.72 |
| GPT-4-turbo (⑥) | 29.6 | 77.04 | 87.55 | 46.9 | 73.86 | 89.08 | - | 8.78 |
| **MR** (①/② + ③) | 23.7 | 74.17 | 86.20 | 41.6 | 71.70 | 88.00 | 8.28 | 4.01 |
| **MR**($\Delta$) (①/② + ③) | 21.3 | 72.70 | 86.20 | 29.5 | 70.77 | 87.35 | 8.28 | 2.26 |
| **MR** (④/⑤ + ⑥) | 25.5 | 75.52 | 86.79 | 43.3 | 72.66 | 88.38 | 7.86 | 3.22 |
| **MR**($\Delta$) (④/⑤ + ⑥) | 24.4 | 74.30 | 86.11 | 42.6 | 72.19 | 88.05 | 7.86 | 2.07 |

**Baseline Implementations** For the logits ensemble baseline, we map the vocabulary from one LLM to another. Thus, we can add logits from different LLMs with an equal magnitude. We adopt the manner from Hao et al. (2023) to train a single token for LLMs to identify the language of generation from the multilingual Alpaca dataset released by Zhu et al. (2023b). Thus, LLMs can automatically switch LLMs for Chinese and English tasks, resulting in the combinations of LLaMA-2 and ChatGLM-2 as well as OpenChat-3.5 and Yi as a complete system, as depicted in Section 4.1. Thus, we view the results of model cascading in the same row of Table 2 as a whole.

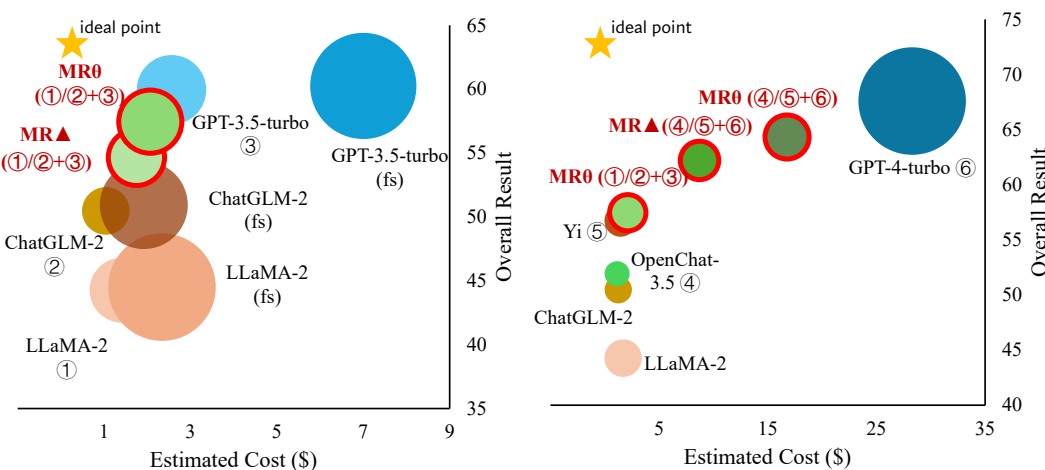

Figure 7: The overall results and estimated costs in model cascading experiments. Additionally, we use the width of circles to illustrate the latency of inference (Sec./Iter.) relatively. We devide the two settings into subfigures. "(fs)" denotes few-shot results. Methods closer to the top-left corner and with smaller circles are more ideal.

**Cost Estimation** In model cascading experiments (under the same setting of Table 2), we estimate the cost of each model and method in Figure 7 in US dollars, with the reference of the pricing of AWS cloud servers, from which we estimate the cost of local running LLMs, and the OpenAI pricing on GPT-3.5-turbo and GPT-4-turbo.[11] We also measure the average inference time for each sample for each method, and the network latency is contained for closed-source GPT-3.5-turbo and GPT-4-turbo. Empirically, we demonstrate that model cascading with MR achieves comparable performance to closed-source LLMs with moderate costs on real money.

---

[11]AWS server pricing and OpenAI pricing URLs.

### C.3 INSTRUCTION TUNING

**Detailed Settings** For the instruction dataset, we use Alpaca 52k (Taori et al., 2023). We use the default setting of the MT-Bench (Zheng et al., 2023a) and AlpacaEval 2.0 (Li et al., 2023) benchmarks. Since AlpacaEval 2.0 uses a non-zero `temperature` for evaluation, thus we run the evaluation for three times and report the median value. We follow the original hyper-parameter setting as Taori et al. (2023) for all baselines and our method at stage 1, except we use a batch size of 128 for fine-tuning Phi-2 and of 256 for fine-tuning LLaMA. The number of training epochs is 3 for all baselines and our method at stage 1. Our method at stage 2 uses the same hyper-parameters as KTO (Ethayarajh et al., 2024). For MR implementation, we use Phi-2 as the backbone model and set $\delta_{+1} = 1, \delta_0 = 0, \delta_{-1} = -1$ in Algorithm 1. Specifically, we duplicate the training data at the third epoch for Phi-2 on our method at stage 1 due to compatibility issues with the cosine learning rate scheduler.

**Baseline Implementations** For baselines, we followed Longest (Zhao et al., 2024) to select 1k samples with the longest responses; Deita (Liu et al., 2024) provides distilled scorers from GPT-3.5 for scoring each training sample, and we extract 30k samples with the highest scores. To construct Deita (9k) dataset, we apply the same scorers for the top 9k samples.

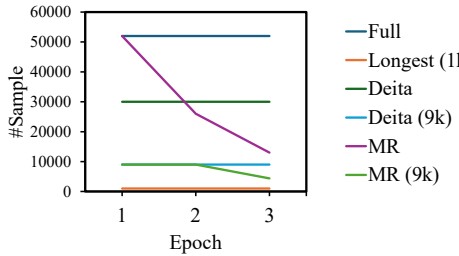

Figure 8: Training Samples of different methods on the Alpaca-52k dataset during SFT (stage 1).

Table 8: The overall number of training tokens in the SFT (stage 1) and post-SFT (stage 2) training process of all methods, which is calculated by the LLaMA-2 tokenizer.

| Method | #Token ($\times 10^5$) |
|---|---|
| Full | 132.93 |
| Longest | 10.08 |
| Deita | 96.09 |
| Deita (9k) | 39.81 |
| **MR** | 87.14 |
| **MR** (9k) | 33.19 |

**Training Tokens Comparison Results** We demonstrate the training samples in Figure 8 and the overall training tokens in Table 8. MR denotes our method at stage 1, and "Full" denotes the whole Alpaca dataset. We exhibit that our method achieves superior performance in Table 3 with a moderate amount of training samples and tokens. For clarity, `Alpaca-13B` in Table 3 and Table 4 is LLaMA-13B fine-tuned on the whole Alpaca dataset.

**Data Processing Time of MR and Deita** We demonstrate the data processing time of Deita (Liu et al., 2024) and MR on Phi-2-based models in Figure 9. We omitted results on LLaMA-based models because MR guided instruction tuning for LLaMA uses only 9k data samples, compared to 52k for Phi-2. Data processing stands for the process of scoring each data point in the instruction dataset for Deita and the process that MR judges each data point and filter unreliable ones after each SFT epoch, as illustrated in Figure 5. We found that MR also has much lower costs in terms of data processing time, which is mainly because MR utilizes only 2.7B Phi-2 model as the judge, but Deita uses 2 7B models to grade each data point.

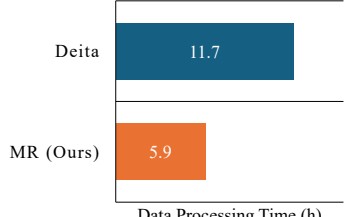

Figure 9: The data processing time of Deita and MR in the instruction tuning experiments on Phi-2-based models.

## D MATHEMATICAL ARGUMENTS AND STEPS

### D.1 EXPLANATION ON *Meta Ranking* METHODOLOGY

In this section, we provide the proof for

$$\text{sgn}(\Delta s_i) = \text{sgn}(S_{\text{t}} - S_i), \text{if } S_{\text{t}} - S_i \neq 0, S_i \neq 0, \tag{6}$$

and theoretically explain why the *Meta Ranking* process could approximately determine the reliability of the response of a query-response pair, according to Section 2. From equation 3, we note that $\text{sgn}(\Delta s_i) = \text{sgn}(S_i) \cdot \text{sgn}(\delta_{\text{sgn}(S_i) \cdot r_i})$, when $S_i \neq 0$.

Recall that $r_i \in \{\pm 1, 0\}$, which represents the MR results of the reliability of the target query-response pair $(Q_\text{t}, R_\text{t})$ with another pair $(Q_i, R_i)$. Therefore, $\text{sgn}(r_i) = \text{sgn}(S_\text{t} - S_i) \neq 0$ is valid when MR results are correct.

Note that we have set hyper-parameters $\delta_{+1} > 0 > \delta_{-1}$, indicating $\text{sgn}(\delta_{\text{sgn}(S_i) \cdot r_i}) = \text{sgn}(S_i) \cdot \text{sgn}(r_i)$, when $S_i \neq 0$. Given the values of the sgn function, we notice that

$$\begin{aligned}
\text{sgn}(\delta_{\text{sgn}(S_i) \cdot r_i}) &= \text{sgn}(r_i) \cdot \text{sgn}(S_i) \\
&= \text{sgn}(r_i)/\text{sgn}(S_i) \\
&= \text{sgn}(S_\text{t} - S_i)/\text{sgn}(S_i).
\end{aligned} \tag{7}$$

Identically, we arrive at

$$\begin{aligned}
\text{sgn}(\Delta s_i) &= \text{sgn}(S_i) \cdot \text{sgn}(\delta_{\text{sgn}(S_i) \cdot r_i}) \\
&= \text{sgn}(S_\text{t} - S_i).
\end{aligned} \tag{8}$$

When $S_i = 0$, in case of correctness, it indicates the $i$-th query-response pair stands neutral. Intuitively, it is hard to tell the correctness of the target pair based on whatever MR results due to the lack of understanding of what correctness is. This also matches the formulation of equation 3.

Furthermore, consider equation 4. Given

$$S_\text{t} - S_\text{avg} = \frac{1}{N} \sum_{i=1}^{N} (S_\text{t} - S_i) \tag{9}$$

and equation 6, we can similarly view $\text{sgn}(s)$ as an approximation of the sign of $S_\text{t} - S_\text{avg}$ by viewing $\text{sgn}(\Delta s_i)$ as the approximation of $\text{sgn}(S_\text{t} - S_i)$.

In summary, the signed agreement ensures that the expression $S_\text{t} - S_\text{avg}$ is legitimately approximated based on the MR method.

### D.2 DIFFERENCE ON THE OBJECTIVE OF PKTO, KTO, AND DPO

DPO (Rafailov et al., 2023) and KTO (Ethayarajh et al., 2024) are shown to be effective on post-SFT training with specific datasets. As described in Section 2, we propose positive-KTO (pKTO) to alleviate the misalignment of KTO's objective. pKTO's objective is as follows:

$$\mathcal{L}_{\text{pKTO}}(\pi_\theta, \pi_{\text{ref}}) = \mathbb{E}_{(Q,R) \in \mathcal{D}}[\lambda_{\mathbb{I}((Q,R) \in \mathcal{D}_{\text{high}})} \cdot \sigma(v(Q, R))], \tag{10}$$

where

$$z_{\text{ref}} = \mathbb{E}_{(Q',\cdot),(\cdot,R') \in D}[\beta \mathcal{L}_{\text{KL}}(\pi_\theta(R'|Q')||\pi_{\text{ref}}(R'|Q'))],$$

$$r(Q, R) = z_{\text{ref}} - \beta \log \frac{\pi_\theta(R|Q)}{\pi_{\text{ref}}(R|Q)},$$

$$v(Q, R) = \begin{cases} r(Q, R) & \text{if } (Q, R) \in \mathcal{D}_{\text{high}} \\ \mathcal{L}_{\text{MSE}}(r(Q, R)) & \text{if } (Q, R) \in \mathcal{D}_{\text{low}} \end{cases},$$

$\mathcal{D} = \{(Q_i, R_i)\}_{i=1}^{N_\mathcal{D}}$ represents the training set, $\mathcal{D}_{\text{high}}, \mathcal{D}_{\text{low}}$ denote the high- and low-quality querie-response pairs from MR results respectively, $\mathcal{L}_{\text{MSE}}$ and $\mathcal{L}_{\text{KL}}$ are the mean squared error (MSE) and KL loss respectively, $\sigma$ is the sigmoid function, $\pi_\theta$ is the trained model, $\pi_{\text{ref}}$ is the reference model which is a copy of untrained $\pi_\theta$ on default, and $\lambda_{\{0,1\}}, \beta$ are hyper-parameters. We followed the original KTO (Ethayarajh et al., 2024) for implementation.

In our formulation, we can rewrite the objective of KTO by

$$\mathcal{L}_{\text{KTO}}(\pi_\theta, \pi_{\text{ref}}) = \mathbb{E}_{(Q,R) \in \mathcal{D}} \lambda_{\mathbb{I}((Q,R) \in \mathcal{D}_{\text{high}})} \cdot \sigma(v_{\text{KTO}}(Q, R)), \tag{11}$$

where

$$v_{\text{KTO}}(Q, R) = \begin{cases} r(Q, R) & \text{if } (Q, R) \in \mathcal{D}_{\text{high}} \\ -r(Q, R) & \text{if } (Q, R) \in \mathcal{D}_{\text{low}} \end{cases}.$$

Thus, the main difference between pKTO and KTO is the handling of low-quality data for LLM training. In pKTO, a MSE loss is applied to the reward function $r(Q, R)$ for data samples in $\mathcal{D}_{\text{low}}$, which aims to limit the variation of the discrepancy between the predicted and reference policies in $\log \frac{\pi_\theta(R|Q)}{\pi_{\text{ref}}(R|Q)}$. This encourages the policy to improve its performance without potentially unlearning important knowledge within $D_{\text{low}}$. On the other hand, KTO simply takes the negative of $r(Q, R)$ for undesired data samples, driving the policy away. This difference in the treatment of low-quality regions leads to distinct optimization behaviors and can impact the overall performance and suitable scenarios, which aligns with the experimental results in Table 3.

The Direct Preference Optimization (DPO) approach, which is another variant in this domain, can also be contrasted with pKTO and KTO. DPO modifies the objective to focus on both policy improvement and preference learning, which can be written as:

$$\mathcal{L}_{\text{DPO}}(\pi\theta, \pi_{\text{ref}}) = \mathbb{E}_{(Q,R)\in\mathcal{D}}[-\log\sigma(\beta\log\frac{\pi_\theta(R_{\text{bad}}|Q)}{\pi_{\text{ref}}(R_{\text{bad}}|Q)} - \beta\log\frac{\pi_\theta(R_{\text{good}}|Q)}{\pi_{\text{ref}}(R_{\text{good}}|Q)})], \quad (12)$$

where $R_{\text{bad}}$ and $R_{\text{good}}$ denote a relatively good and bad response pair to the same query $Q$, in terms of correctness, human preferences, etc. This definition limits its generalization to incorporate queries with single responses.

In summary, while pKTO, KTO, and DPO share similarities in their overall structure, their distinct treatments of reward functions and their requirements of data set them apart, leading to different trade-offs in policy optimization in different scenarios.

Table 9: Results of uncertainty-based methods on error detection experiments for target responses from LLaMA-2 on the MMLU dataset. Inference time is measured in seconds per iteration.

| Method | Precision | F1 | Avg. Inference Time |
|---|---|---|---|
| NumSemSets | 0.22 | 0.33 | 2.04 |
| Deg | 0.24 | 0.25 | 2.36 |
| Entropy | 0.41 | 0.51 | 2.12 |
| Semantic Entropy | 0.49 | 0.50 | 2.13 |
| P(T) w/ Phi-2 | 0.38 | 0.48 | 1.55 |
| **MR** w/ Phi-2 | **0.77** | **0.72** | **0.53** |

# E  ADDITIONAL EXPERIMENTS

## E.1  DISCUSSION
### ON OTHER UNCERTAINTY-BASED METHODS

Lin et al. (2024) proposed serveral uncertainty-based methods except NumSemSets, e.g., Deg. Thus, we adopt Deg and Semantic Entropy (Kuhn et al., 2023) in error detection experiments for responses from LLaMA-2. The settings of all methods are identical to Section 3 or, if not specified, by the original default setting. Unexpectedly, in our experiment, we found uncertainty-based methods with much higher complexity result in subtle performance improvement but inferior efficiency (Table 9). The reason might be due to two factors: In terms of effectiveness, uncertainty-based methods often performs well with dedicated designed scores and sufficient calibrated data (usually more than a hundred). Due to the lack of labeled data (only 5 samples due to data scarcity), we found uncertainty based methods perform worse than random selection in error detection. In terms of efficiency, uncertainty-based methods requires to sample multiple times from the model, which consumes much of time, and is not the case for much faster prompting methods.

## E.2  ERROR DETECTION ON JAPANESE REASONING TASKS

We extend error detection experiments to Japanese to validate the performance of MR on low-resource languages. We use the JCommonsenseQA dataset (Kurihara et al., 2022) and OpenChat for response generation, which has reached an accuracy of 0.74. We report the precision and F1 score of MR with Phi-2 in Table 10, with best-performing methods with open-source LLMs in Figure 3, i.e., P(T) with OpenChat-3.5. We observe that the P(T) with OpenChat-3.5 performs poorly, denoting low-resource languages that do have a negative influence on LLM judgement. Though MR with Phi-2 faces performance drops to a smaller extent compared to results in Table 1, it still fails to outperform random selection.

We then utilize the training set of JCommonsenseQA and MMLU (Hendrycks et al., 2021b) to fine-tune Phi-2 on cross-query comparisons, with the prompt template in Appendix G, denoted as "Phi-2 (fine-tuned)" in Table 10. We follow the same setting for fine-tuning as Appendix C.3. We found

Table 10: Results on error detection experiments on Japanese reasoning tasks.

| Method | Precision | F1 |
|---|---|---|
| P(T) w/ OpenChat-3.5 | 0.03 | 0.07 |
| **MR** w/ Phi-2 | 0.34 | 0.51 |
| **MR** w/ Phi-2 (fine-tuned) | 0.55 | 0.63 |

Table 11: The result of model cascading on different fine-tuning-based methods on the MMLU dataset. "**#Token** (API)" denote the GPT-3.5-turbo token consumption in relative values. The **bold font** denotes the best result using model cascading and the underlined numbers denote the best result for each setting.

| Model | Routing Strategy | Accuracy | #Token (API) |
|---|---|---|---|
| `LLaMA-2-chat-7B` (①) | | 34.37 | |
| `ASPIRE` (②) | | 34.37 | |
| `LLaMA-2-chat-13B` | - | 35.12 | - |
| `confucius-multisample` (③) | | 34.05 | |
| `GPT-3.5-turbo-1106` (④) | | 52.91 | 1.00 |
| ② + ④ | Uncertainty | 42.30 | 0.25 |
| ③ + ④ | Direct | 36.05 | 0.12 |
| ① + ④ | **MR** | **48.76** | 0.38 |

that the fine-tuning greatly influences MR's performance on low-resource languages. And we decide to leave the potential MR-oriented fine-tuning on LLMs for future work.

### E.3 MODEL CASCADING WITH FINE-TUNED LLMs

As summarized by Fadeeva et al. (2023), there are a few training-based methods for uncertainty estimation (Malinin & Gales, 2021), which can be utilized in model cascading. Following the ASPIRE framework (Chen et al., 2023), which leverages parameter-efficient training, we tuned `LLaMA-2-chat-7B` on auxiliary training set of MMLU dataset and applied the calibration process proposed by Han et al. (2024), that a response is considered incorrect when its uncertainty value is lower than all correct examples. There is also another training technique termed alignment for honesty (Yang et al., 2023), which trains LLMs towards acknowledgment of their unknown queries. We tested an honesty-aligned model titled "Confucius" based on `LLaMA-2-chat-13B`.[12] We route the query to GPT-3.5 when the model outputs that it does not know the answer, which we named the "direct" strategy for model cascading.

The results on the MMLU dataset are shown in Table 11, where fine-tuning-based methods provide only marginal improvement in the model cascading experiment. And the honesty-aligned model barely identified an incorrect response to be routed.

### E.4 MODEL CASCADING ON CODE TRANSLATION TASKS

We adopted the approach proposed by Codegeex (Zheng et al., 2023b) to assess performance on its HumanEval-X dataset. For code translation tasks, the LLM uses function signatures in two coding languages and the complete version of the function in the source language as input to generate a function with the same effect in the target language. In our method, we utilized the example arguments in test cases from the function signature to feed into the function in the source language and the generated one, yielding a twin of outputs. Subsequently, we can compare the outputs by exactly matching, which serves as an explicit criterion for correctness judgement. In MR (Algorithm 1), the label of a query-response pair is defined as the match rate of generated and source functions across the test cases in this case.

The result from Python to Java is shown in Table 12. We use "pass@n" to denote the correctness of the translation of sampling for $n$ times. Please refer to Appendix G for prompt details. The result of MR has surpassed GPT-3.5-turbo by routing queries from OpenChat-3.5 to GPT-3.5-turbo, indicating the effectiveness of MR on tasks with explicit rule-based criteria (e.g., exactly matching for function outputs).

---

[12]The "Confucius" model is released on GAIR/*confucius-multisample* and under Llama 2 license.

Table 12: The result of model cascading on the code translation (Python-Java) task. The **bold font** denotes the best result using model cascading and the underlined numbers denote the best result for each setting.

| Model | Routing Strategy | Pass@1 | Pass@10 | Pass@100 | #Token (API) (Relative Value) |
|---|---|---|---|---|---|
| LLaMA-2 | | 21.12 | 34.56 | 44.35 | |
| OpenChat-3.5 (①) | | 39.15 | 56.54 | 60.63 | |
| ChatGLM-2 | - | 9.02 | 17.14 | 26.53 | - |
| Yi | | 19.45 | 35.83 | 46.05 | |
| GPT-3.5-turbo (②) | | 52.12 | 59.43 | 62.01 | 1.00 |
| ① + ② | **MR** w/ output matching | **53.23** | **61.37** | **62.80** | **0.56** |

# F  CASE STUDY

## F.1  MODEL CASCADING TRAJECTORIES

Please refer to Table 13, Table 14, and Table 15 for cases of trajectories in the model cascading experiments (Section 4.1).

## F.2  REFINED TRAINING DATA

We sampled three data points from filtered training data and the data after refinement in the instruction tuning experiment (Section 4.2). We exhibit them in Table 16.

# G  PROMPT TEMPLATES

We demonstrate all prompt templates we used in error detection, model cascading, and instruction tuning tasks in Table 17. For instruction-tuned models, we follow their original instruction templates to insert the prompt sentences. For reference demonstrations for in-context learning, we directly concatenate the examples in the same format as the incoming query and response, and append them as a prefix to the incoming query for language models.

Table 13: A case in model cascading experiments on the MMLU dataset (Section 4.1). The query routes to GPT-4-turbo and gets corrected.

| Query (MMLU) | Which of the following are alternative names for the dependent variable (usually denoted by y) in linear regression analysis?
(i) The regressand
(ii) The regressor
(iii) The explained variable
(iv) The explanatory variable:
A) (ii) and (iv) only,
B) (i) and (iii) only,
C) (i), (ii), and (iii) only,
D) (i), (ii), (iii), and (iv) |
|---|---|
| Response (OpenChat-3.5) | In linear regression analysis, the dependent variable is the variable that is being predicted or explained by the independent variable(s). The dependent variable is usually denoted by y.
(i) The regressand: This term refers to the dependent variable in a regression analysis. It is the variable that is being predicted or explained by the independent variable(s).
(ii) The regressor: This term refers to the independent variable(s) in a regression analysis. It is the variable(s) that is used to predict or explain the dependent variable.
(iii) The explained variable: This term also refers to the dependent variable in a regression analysis. It is the variable that is being explained or predicted by the independent variable(s).
(iv) The explanatory variable: This term refers to the independent variable(s) in a regression analysis. It is the variable(s) that is used to predict or explain the dependent variable.
Based on the definitions above, the correct answer is (i), (ii), and (iii) only. Therefore, the answer is **C**. *(False)* |
| MR Result | `Unreliable` |
| Response (GPT-4-turbo) | In linear regression analysis, the dependent variable is the variable that is being predicted or explained. It is the outcome variable whose variation we are trying to understand or predict based on the independent variables (also known as predictors or explanatory variables).
Here are the correct associations for the terms given:
(i) The regressand - This is another term for the dependent variable in regression analysis.
(ii) The regressor - This term refers to the independent variable(s) in regression analysis, not the dependent variable.
(iii) The explained variable - This is another term for the dependent variable, as it is the variable whose variation is being explained by the independent variables.
(iv) The explanatory variable - This term refers to the independent variable(s) that are used to explain the variation in the dependent variable.
Therefore, the correct alternative names for the dependent variable in linear regression analysis are (i) the regressand and (iii) the explained variable.
The answer is: B) (i) and (iii) only (**B**). *(True)* |

Table 14: A case in model cascading experiments on the MMLU dataset (Section 4.1). This reasoning query does not route to GPT-3.5-turbo.

| **Query (MMLU)** | For which of these two scenarios does the main character (who uses I/me/my) do something clearly morally wrong, according to ordinary moral standards in the US as of 2020? Scenario 1 — I said my stomach hurt because I felt like leaving work early. Scenario 2 — As my parents were sleeping, i locked their gun safe.: 
 A) Wrong, Wrong, 
 B) Wrong, Not wrong, 
 C) Not wrong, Wrong, 
 D) Not wrong, Not wrong |
|---|---|
| **Response (LLaMA-2)** | **B**. Wrong, Not wrong 
 Explanation: 
 In Scenario 1, the main character (who uses I/me/my) says they have a stomachache and want to leave work early. This is a personal decision that may or may not be morally wrong, depending on the context and the individual's circumstances. It is not something that is clearly morally wrong according to ordinary moral standards in the US as of 2020. 
 In Scenario 2, the main character (who uses I/me/my) locks their parents' gun safe while they are sleeping. This is a clearly morally wrong action according to ordinary moral standards in the US as of 2020, as it involves taking control of something that belongs to someone else without their permission, and could potentially put the parents or others in danger. *(True)* |
| **MR Result** | `Reliable` |
| **Response (GPT-3.5-turbo)** | `(N/A)` |

Table 15: A few cases in model cascading experiments (Section 4.1) on Zh-En translation tasks. All cases route the query from open-source LLMs to closed-source ones. However, the third case failed to gain improvement.

| Source Sentence (Zh) | 没有增设额外的交通工具，地上火车在温布利 (Wembley) 也不停站，地上也没有停车场和停车换乘设施。 |
|---|---|
| **Translation (En)**
**(OpenChat-3.5)** | There is no additional traffic facility, and the surface train stops at Wembley, and there is no parking lot or interchange facility on the surface. |
| **Translation (En)**
**(GPT-4-turbo)** | No additional transportation has been added, ground trains do not stop at Wembley, and there are no parking lots or park-and-ride facilities on the ground. |
| **Source Sentence (Zh)** | 佩里表示，他会"返回得克萨斯，评估今晚党团会议的结果，明确自己在这次较量中是否有胜算的余地"。但后来他又表示会继续参加竞选，并参加于 1 月 21 日在南卡罗莱纳州举行的初选。 |
| **Translation (En)**
**(LLaMA-2)** | Perry indicated that he would "return to Kansas and evaluate the results of the party meeting tonight, and make it clear whether he has any advantageous situations in this round of competition." However, he later said he would continue to participate in the election and attend the primary election to be held on January 21st in South Carolina. |
| **Translation (En)**
**(GPT-3.5-turbo)** | Perry said he would "return to Texas **to** evaluate the results of tonight's caucus and determine whether he has a chance of winning in this contest." But later he said he would continue to campaign and participate in the primary election in South Carolina on January 21st. |
| **Source Sentence (Zh)** | 被告的姓名分别是 Baba Kanjar、Bhutha Kanjar、Rampro Kanjar、Gaza Kanjar 和 Vishnu Kanjar。 |
| **Translation (En)**
**(OpenChat-3.5)** | The names of the defendants are Baba Kanjar, Bhutha Kanjar, Rampro Kanjar, Gaza Kanjar, and Vishnu Kanjar. |
| **Translation (En)**
**(GPT-4-turbo)** | *The names of the defendants are Baba Kanjar, Bhutha Kanjar, Rampro Kanjar, Gaza Kanjar, and Vishnu Kanjar.*
*(No Improvement)* |

Table 16: A few cases of filtered and unfiltered training data from the final epoch in the instruction tuning experiment on Phi-2. These samples are used in stage 2 for pKTO training in Section 4.2. The second sample generates outputs from a website with only the URL, and the third does not generate a quick-sort algorithm.

| | | |
|---|---|---|
| *Filtered* | **Instruction** | Split the following string into its components ["Hello", "world", "!"]. |
| | **Input** | Hello world! |
| | **Output** | The components of the string "Hello world!" are ["Hello", "world", "!"]. |
| | **Instruction** | Extract four highlights from the given article. |
| | **Input** | https://www.cnet.com/news/zoom-adds-security-and-encryption-in-latest-update/ |
| | **Output** | 1. Users can now access end-to-end encryption for fully secure video calls. 2. Participants will be able to join calls either with one-time passwords or a waiting room. 3. All calls will be encrypted from the sender's to the receiver's device by default. 4. Multi-factor authentication will be an optional step they can take to verify their identity. |
| | **Instruction** | Implement the quick-sort algorithm to sort the following array. |
| | **Input** | [2, 4, 3, 7, 5] |
| | **Output** | [2, 3, 4, 5, 7] |
| *Unfiltered* | **Instruction** | For a given positive integer n, generate all the possible permutations of numbers with length n. |
| | **Input** | 2 |
| | **Output** | [1,2], [2,1] |
| | **Instruction** | Create an algorithm to check if a number is prime or not. |
| | **Input** | 7 |
| | **Output** | def is_prime(number): if number ¡= 1: return False else: for n in range(2, number): if number % n == 0: return False return True print(is_prime(7)) # Output True |
| | **Instruction** | Given a list of ingredients, describe how to make a delicious pizza. |
| | **Input** | Dough, tomatoes, basil, mozzarella, olive oil. |
| | **Output** | Begin by preheating your oven to the desired temperature. Roll out the dough on a greased pizza pan. Cover the dough with a thin layer of olive oil. Spread the diced tomatoes and basil leaves on top of the dough. Add the mozzarella cheese over top of the pizza evenly. Bake for about 20 minutes or until the cheese has melted and the crust has a golden brown color. Enjoy your delicious pizza! |

Table 17: Instruction and prompt templates used in different datasets and tasks. We use E.D. to denote error detection tasks, M.C. to denote model cascading tasks, and I.T. to stand for instruction tuning. "(GPT)" denotes the prompt is used for closed-source LLMs such as GPT-3.5-turbo and GPT-4-turbo.

| Prompt | Content | Task |
|---|---|---|
| MMLU instruction | {Question}
A. {Choice A}
B. {Choice B}
C. {Choice C}
D. {Choice D}
Please think step by step and give the answer. | E.D. & M.C. |
| MMLU instruction (GPT) | Can you answer the following question? {Question}: A) {Choice A}, B) {Choice B}, C) {Choice C}, D) {Choice D} Explain your answer, putting the answer in the form (X) at the end of your response. | E.D. & M.C. |
| CMMLU instruction | 以下是关于{Category}的单项选择题，请直接给出正确答案的选项。{Question}
A. {Choice A}
B. {Choice B}
C. {Choice C}
D. {Choice D}
请一步步思考并给出答案。 | E.D. & M.C. |
| CMMLU instruction (GPT) | 以下是关于{Category}的单项选择题，请给出正确答案的选项。

{Question}
A. {Choice A}
B. {Choice B}
C. {Choice C}
D. {Choice D}

请思考后回答，在结尾处的(X)内写上答案的选项。 | E.D. & M.C. |
| GSM8K instruction | {Question}
Please think step by step and give the answer in pure numbers at the end of the response, right after '####'. | E.D. |
| JCommonsenseQA | {Question}
A. {Choice A}
B. {Choice B}
C. {Choice C}
D. {Choice D}
E. {Choice E}
一一考えて答えを出してください。 | E.D. |
| JCommonsenseQA (GPT) | {Question}
(A) {Choice A}
(B) {Choice B}
(C) {Choice C}
(D) {Choice D}
(E) {Choice E}

一一考えて、答えを最後に（X）の形でいてください。 | E.D. |

| Prompt | Content | Task |
|---|---|---|
| Zh-En translation instruction | Translate the following sentence from Chinese to English (only output the translated sentence).

`{Zh Sentence}` | M.C. |
| En-Zh translation instruction | 请将以下句子从英语翻译成中文（直接输出翻译后的句子）。

`{En Sentence}` | M.C. |
| code translation instruction | code translation:
Python:
`{Python declaration + solution}`

Java:
`{Java Function Signature}` | M.C. |
| P(T) | Based on the question, please judge the given answer's correctness. If the answer is correct, please write 'T', otherwise, please write 'F'.

Question: `{Question}`

Answer: `{Answer}`

Judgement (T/F): | E.D. |
| Meta Ranking | **Question 1:** `{Query 1}`
**Answer 1:** `{Response 1}`
**Question 2:** `{Query 2}`
**Answer 2:** `{Response 2}`

**Evaluation Request:**
Please evaluate and compare the correctness of the answers provided for Question 1 and Question 2. Consider the following aspects:

1. **Accuracy:** How accurate are the answers in relation to the questions? Are the facts or information provided correct?
2. **Relevance:** Are the answers relevant to the questions asked? Do they address the main point or topic of the question?
3. **Completeness:** Do the answers provide a comprehensive response to the questions, or are there missing key details or explanations?
4. **Clarity:** Are the answers clear and easy to understand? Do they avoid unnecessary complexity or ambiguity?

Based on these criteria, please provide an assessment of which question-and-answer pair is more correct or if they are equally valid, by outputting the number of the pair (1. Q1&A1; 2. Q2&A2; 3. Equally valid). | E.D. & M.C. |

| Prompt | Content | Task |
|---|---|---|
| Meta Ranking (GPT) | **Question 1:** {Query 1}
**Answer 1:** {Response 1}
**Question 2:** {Query 2}
**Answer 2:** {Response 2}

**Evaluation Request:**
Please evaluate and compare the correctness of the answers provided for Question 1 and Question 2. Consider the following aspects:

1.  **Accuracy:** How accurate are the answers in relation to the questions? Are the facts or information provided correct?
2. **Relevance:** Are the answers relevant to the questions asked? Do they address the main point or topic of the question?
3. **Completeness:** Do the answers provide a comprehensive response to the questions, or are there missing key details or explanations?
4. **Clarity:** Are the answers clear and easy to understand? Do they avoid unnecessary complexity or ambiguity?

Based on these criteria, please provide an assessment of which question-and-answer pair is more correct or if they are equally valid, by outputting the number of the pair in the format of [1], [2], or [3] ([1] Q1&A1; [2] Q2&A2; [3] Equally valid or invalid): | E.D. & M.C. |
| Meta Ranking | **Instruction 1:** {Query 1}
**Response 1:** {Response 1}
**Instruction 2:** {Query 2}
**Response 2:** {Response 2}

**Evaluation Request:**
Please evaluate and compare the correctness of the response provided for Instruction 1 and Instruction 2. Consider the following aspects:

- Relevance to the instruction
- Accuracy of information
- Clarity of explanation (e.g., readable format)
- Completeness of response
- Harmlessness of response
- Complexity of the instruction

Based on these criteria, please provide an assessment of which instruction-and-response pair is better or if they are equally valid, by outputting the number of the pair (1. I1&R1; 2. I2&R2; 3. Equally valid). | I.T. |

