# OpenReview forum: "Enabling Weak LLMs to Judge Response Reliability via Meta Ranking"
_ICLR.cc/2025/Conference — Submitted to ICLR 2025_

### Official Review · Reviewer_fbvQ · 2024-10-16

**Soundness:** 3
**Presentation:** 3
**Contribution:** 2
**Rating:** 3
**Confidence:** 3

**Summary:**

The innovation of this article is reflected in the fact that it proposes a novel Meta Ranking method that can effectively solve the reliability problem of weak LLMs in response evaluation, and demonstrates its applications through practical application scenarios such as model cascading and instruction fine-tuning. application potential.

**Strengths:**

1. MR uses a weighted voting system that compares target and reference pairs, incorporating reliability scores for accurate response assessment.
2. MR outperforms traditional uncertainty methods (e.g., entropy, confidence) with better generalization and robustness across tasks and datasets.
3. Cross-query comparisons reduce overconfidence in LLMs, improving accuracy, especially in weak models.

**Weaknesses:**

1. While there are theoretical innovations, they lack sufficient depth. The paper does not clearly explain why the MR voting mechanism leads to improved performance and judgment efficiency.
2. Although experiments demonstrate the method's effectiveness, the overall scale is limited. There are not enough experiments exploring how different MR configurations and settings impact performance improvement.
3. Despite introducing two meaningful applications, their content is insufficient to fully enrich the second half of the paper and lacks substantial innovation. More exploration is needed to highlight their significance and novelty.

**Questions:**

In Figure 3, after the introduction of MR, can the inference time of the same large language model drop so much just because the input text length becomes shorter?

---

### Official Review · Reviewer_YyP2 · 2024-11-04

**Soundness:** 2
**Presentation:** 2
**Contribution:** 2
**Rating:** 5
**Confidence:** 2

**Summary:**

This paper proposes Meta Ranking, a method to evaluate the reliability of LLM responses with weak LLMs. Before this work, there are 2 major paradigms to evaluate LLM reliability:
- Zero shot: given $(Q, R)$ pairs, LLM evaluate and generate evaluation scores. Weak LLM might not be able to provide correct judgment.
- In-context-learning: given $(Q, R)$ pairs, plus references pair $(Q_r, R_r)$, concatenate the reference pairs along with the query and response pairs as input, LLM gives the judge. The order of concatenation matters.

The proposed method uses each reference pair separately without concatenation to reduce the effect of permutation. The weak LLM takes each $(Q, R, Q_r, R_r)$ pair as input, and generates a score. For $N$ reference pairs, the LLM will run $N$ times of inference to get $N$ scores. Then voting-based aggregation is used to generate the final evaluation score for this pair.

The proposed method is evaluated with major benchmark datasets in terms of micro precision scores to show improvement, and tested in 2 applications: model cascading and instruction tuning.

**Strengths:**

- The proposed method is simple and easy to understand.
- Solid evaluation to show the improvement in model reliability evaluation, and through downstream applications.

**Weaknesses:**

- Novelty seems to be limited. Though it's called meta-ranking, the actual method can be viewed as a special case of ICL: instead of concatenating $N$ reference pairs in one query and send to LLM, each reference pair + query and response to be evaluated are sent to LLM.
- The definition of "Reliability" is vague. The definition in this paper is "attributes such as correctness and quality as required by the context." As it's not a strict definition, it's hard to evaluate the results and understand the improvement.
- Efficiency concerns. The proposed method is not scalable. If there are $N$ references, then $N$ times of inference is needed. Compared with the original ICL method, it increases the computation complexity from $O(1)$ to $O(N)$. It's mentioned in the paper that "In practice, N is usually small due to efficiency and the limited labeled data.", but this is a strong assumption and it might not always be the case.

**Questions:**

- Is it possible to give a more rigorous definition of reliability? This is the core metric to be evaluated in this work, it would be very helpful.

---

### Official Review · Reviewer_ocux · 2024-11-06

**Soundness:** 3
**Presentation:** 2
**Contribution:** 2
**Rating:** 3
**Confidence:** 4

**Summary:**

This paper proposes a novel cross-query-comparison-based method called Meta Ranking (MR). MR assesses reliability by pairwise ranking the target query-response pair with multiple reference query-response pairs. The method is highly effective in error detection and can enhance strong models in model cascading and instruction tuning.

**Strengths:**

* The proposed method is novel and demonstrates superior performance in error detection, model cascading and instruction tuning.

* The method is effective even with weaker LLMs, showcasing the potential for weak-to-strong generalization.

**Weaknesses:**

* Section 4 is poorly written and difficult to follow.

* Even with a weaker model, the proposed method significantly increases the inference cost. Therefore, it should be compared with other inference-scaling methods such as self-consistency.

* The paper should include more recent models such as LLaMA-3 and GPT-4o-(mini).

* Table 1
  - It’s weird that Phi-2 obtains zero accuracy on GSM8K and performs close to random guessing (~25%) on MMLU and CMMLU.
  - It should at least include the task performance of the base models for comparison, i.e. LLaMA-2, ChatGLM-2 and Yi.
  - What does "Random Selection" mean?
  - If MR uses 5 few-shot samples (some are correct and some are incorrect), do you also include 5 few-shot samples (same questions, but all responses are correct) for the baselines (TP of Phi-2, P(T) w/ OpenChat-3.5, etc)? Otherwise, it’s difficult to tell if the benefit is from MR or the addition of the 5 few-shot samples.

* In Figure 3, why P(T) w/Phi-2 has a much higher inference time than MR w/Phi-2? MR requires the model to compare N=5 pairs and P(T) just directly output the correctness of the query.

* In Figure 4, the performance increases as you increase the number of reference pairs. What the curve would look like when it is greater than 5?

**Questions:**

Please see weakness.

---

### Meta-Review · Area_Chair_UMWa · 2024-12-09

**Metareview:**

The submission proposes a method to evaluate the reliability of LLM responses with weak LLMs.  Reviewers were unanimous that the submission is not of sufficient quality for acceptance to ICLR.  Concerns included insufficient depth of the contribution, increased inference cost, and writing quality.  The authors did not provide a rebuttal.

**Additional Comments On Reviewer Discussion:**

The authors did not provide a rebuttal, and the reviewers were unanimous that the submission should not be accepted.

---

### Decision · Program_Chairs · 2025-01-22

Reject